# Nitrogen Journey in Plants: From Uptake to Metabolism, Stress Response, and Microbe Interaction

**DOI:** 10.3390/biom13101443

**Published:** 2023-09-25

**Authors:** Omar Zayed, Omar A. Hewedy, Ali Abdelmoteleb, Mohammed Ali, Mohamed S. Youssef, Ahmed F. Roumia, Danelle Seymour, Ze-Chun Yuan

**Affiliations:** 1Department of Botany and Plant Sciences, University of California Riverside, Riverside, CA 9250, USA; omarz@ucr.edu; 2Genetics Department, Faculty of Agriculture, Menoufia University, Shebin El-Kom 32511, Egypt; hewedy.omar@gmail.com; 3Department of Plant Agriculture, University of Guelph, 50 Stone Road East, Guelph, ON N1G 2W1, Canada; 4Botany Department, Faculty of Agriculture, Menoufia University, Shebin El-Kom 32511, Egypt; aly_gera@yahoo.com; 5Maryout Research Station, Genetic Resources Department, Desert Research Center, 1 Mathaf El-Matarya St., El-Matareya, Cairo 11753, Egypt; mohammedalidrc@gmail.com; 6Botany and Microbiology Department, Faculty of Science, Kafrelsheikh University, Kafrelsheikh 33516, Egypt; m.s.gad@sci.kfs.edu.eg; 7Department of Plant Science, University of Manitoba, Winnipeg, MB R3T 2N2, Canada; 8Department of Agricultural Biochemistry, Faculty of Agriculture, Menoufia University, Shibin El-Kom 32514, Egypt; ahmed.fouad95@agr.menofia.edu.eg; 9Agriculture and Agri-Food Canada, 1391 Sandford Street, London, ON N5V 4T3, Canada; 10Department of Microbiology and Immunology, The University of Western Ontario, 1151 Richmond Street, London, ON N6A 5B7, Canada

**Keywords:** beneficial microbes, nitrogen metabolism, nitrogen assimilation, nitrate transporters, nitrite transporters, ammonium transporters

## Abstract

Plants uptake and assimilate nitrogen from the soil in the form of nitrate, ammonium ions, and available amino acids from organic sources. Plant nitrate and ammonium transporters are responsible for nitrate and ammonium translocation from the soil into the roots. The unique structure of these transporters determines the specificity of each transporter, and structural analyses reveal the mechanisms by which these transporters function. Following absorption, the nitrogen metabolism pathway incorporates the nitrogen into organic compounds via glutamine synthetase and glutamate synthase that convert ammonium ions into glutamine and glutamate. Different isoforms of glutamine synthetase and glutamate synthase exist, enabling plants to fine-tune nitrogen metabolism based on environmental cues. Under stressful conditions, nitric oxide has been found to enhance plant survival under drought stress. Furthermore, the interaction between salinity stress and nitrogen availability in plants has been studied, with nitric oxide identified as a potential mediator of responses to salt stress. Conversely, excessive use of nitrate fertilizers can lead to health and environmental issues. Therefore, alternative strategies, such as establishing nitrogen fixation in plants through diazotrophic microbiota, have been explored to reduce reliance on synthetic fertilizers. Ultimately, genomics can identify new genes related to nitrogen fixation, which could be harnessed to improve plant productivity.

## 1. Introduction

Soil erosion, compaction, acidification, and contamination are significant factors causing soil degradation, affecting 50% of agricultural land and food production [1,2,3]. Consequently, 75 billion tons of fertile soil are lost worldwide due to degradation annually [4]. Moreover, massive amounts of synthetic agro inputs to fertilize crops used by farmers harm environmental health [5,6]. In this sense, applying nitrogen (N) fertilizers has increased sharply by 7.4 times compared with crop productivity, which has only increased by 2.4 times, indicating that crops have been reduced in their ability to use N efficiently [7], increasing food insecurity [8,9]. Therefore, innovative and sustainable strategies in the agro-biotechnological sector have received more attention from botanical scientists worldwide over the last century to solve food security challenges and develop environmentally friendly products [10]. Throughout time, microbial diversity and biomass have been assessed as partners in the plant–microbe association in maintaining soil multifunctionality [11].

N is a vital macronutrient for plants and a crucial component of amino acids that serve as the building blocks of enzymes and proteins in plants. Additionally, N is a part of the chlorophyll molecule, an essential factor in photosynthesis for absorbing sunlight energy, promoting plant growth and grain yield. Even the roots contain this essential element because proteins and enzymes control water and nutrient intake [12]. Recently, Zhang et al. (2022) found that N can significantly impact flowering time [13]. In addition to being a crucial nutrient for plant growth and development, N has a strong correlation with several abiotic stress responses [14]. Moreover, N has a leading role in plant adaptation to the deficiency of macro- and micronutrients. For macronutrients, the processes of N and phosphorus (P) uptake interact together, creating a nutritional balance under fluctuating availability of nutrients [15], while potassium (K) and nitrate (NO_3_^−^) translocation and intake are positively connected as well [16]. The plant nitrate transporters (NRTs) regulate the K translocation [17]. Regarding micronutrients, ammonium (NH_4_^+^) application promotes iron (Fe) uptake [18]; in contrast, NO_3_^−^ triggers the chlorosis symptoms of Fe deficiency [19]. Sulfur (S), a crucial component of enzymes involved in N metabolism [20], is correlated with N uptake [21]. Alternatively, a reverse relationship between molybdenum (M) and N has been noticed [22]. In addition, through the mechanism of anion-cation balance, approximately 90% of stress responses are influenced by NO_3_^−^ uptake, which promotes the synergetic transfer of cations, including K^+^, sodium (Na^+^), cadmium (Cd^2+^), and zinc (Zn^2+^) while inhibiting the absorption of anions like chloride (Cl^−^) and sulfate (SO_4_^−^) [14]. Under drought stress, NO_3_^−^ and NH_4_^+^ concentrations have unique impacts on plant performance. While the usage of NH_4_^+^ reduces the effects of drought on plant development, NO_3_^−^ application has a reverse influence [23].

In this review, we aim to provide a detailed story of the N journey starting from its uptake from soil either in NO_3_^−^ or NH_4_^+^ form, followed by a structural investigation of the transporters of these two forms. Upon reaching their destination, NO_3_^−^ and NH_4_^+^ are exposed to various metabolic pathways (e.g., oxidation and denitrification). Next, N assimilation occurs under the intersection with glutamine synthetase (GS) and glutamate synthase or glutamine 2-oxoglutarate aminotransferase (GOGAT). Then, a comprehensive survey of the role of N in plant adaptation/tolerance to abiotic stresses, either drought or stress, is discussed. Finally, the detrimental effects of nitrogen mineral fertilization and the possibility of microbial application and genomic approaches are summarized.

## 2. Nitrogen Uptake

N is an essential macronutrient for plants, and it can be acquired from the soil in inorganic forms (NO_3_^−^ and NH_4_^+^), which are transported across the root plasma membrane by different families of transporters. Under normal soil conditions, N is mainly available in the form of NO_3_^−^, and four families of transporters mediate NO_3_^−^ uptake, namely NRT1, NRT2, chloride channel (CLC-1), and slow anion channel-associated 1 homolog 3 (SLAC1/SLAH). These transporters have distinct characteristics in affinity, capacity, regulation, and localization [24]. NH_4_^+^ uptake is facilitated by ammonium transporters (AMTs), high-affinity transporters expressed mainly in the root hairs and epidermis [14]. NH_4_^+^ is the dominant form of N in flooded or acidic soils, and AMTs-mediated acquisition is crucial for the N demand of plants growing in such conditions [25]. Besides inorganic N, plants can also absorb organic N in the form of amino acids (AAs), which are abundant in soils that receive organic amendments, such as manure or compost [26]. Several AAs transporters have been identified in plant roots, including amino acid permease (AAP1 and AAP5), proline transporter (ProT2), and lysine and histidine transporters (LHT1 and LHT6) [27,28]. These transporters have different substrate specificities and expression patterns contributing to the uptake of a range of amino acids from the soil solution [27,29]

## 3. Nitrogen Transporters

### 3.1. Plant Nitrate Transporters

Plants acquire NO_3_^−^ from the soil through various transporters that belong to different families [30]. The NRT1 and NRT2 families are the primary transporters involved in NO_3_^−^ uptake, and they have distinctive roles in plant growth and seed development. The NRT1 family is a large and diverse group of nitrate peptide transporter family (NPF). There are 53 NRT1 genes in *Arabidopsis* and 93 in rice [31]. These genes can be classified into 8–10 subfamilies; NRT1.1 and NRT1.2 are responsible for NO_3_^−^ root uptake. The NRT1 family can transport NO_3_^−^ and other substrates such as hormones, nitrite, amino acids, peptides, chloride, glucosinolates, and jasmonate-isoleucine [32]. The NRT2 family is a smaller, more specific group of transporters expressed mainly under low NO_3_^−^ conditions. In *Arabidopsis*, four NRT2 transporters (NRT2.1, NRT2.2, NRT2.4, and NRT2.5) function in nitrate influx and account for 95% of nitrate uptake under low NO_3_^−^ concentrations [33]. However, NRT2.1 and NRT2.2 are significant members of the NRT2 family for nitrate uptake. The NRT2 family has different spatial and temporal expression patterns, and they transport NO_3_^−^ from different sources to different tissues. For example, NRT2.4 and NRT2.5 absorb NO_3_^−^ from the soil root hairs, while NRT2.1 and NRT2.2 transport it from the apoplast to the cortex and endodermis cells. Moreover, NRT2.5 is induced by long-term starvation and enhances NO_3_^−^ uptake from both shoots and roots of mature plants [33,34]. Furthermore, another transporter family called NRT3 is also involved in NO_3_^−^ transport in plants. The NRT3 family consists of two members: NRT3.1 and NRT3.2. These transporters form a complex with NRT2 transporters and regulate their activity and stability [35].

NO_3_^−^ transceptor (transporter/receptor) NRT1.1 regulates the expression levels of many NO_3_^−^ assimilation pathway genes by sensing the external NO_3_^−^ concentration and modulating the root growth accordingly [36,37]. NO_3_^−^ transporters are regulated through phosphorylation, mediated by calcium-dependent calcineurin B-like (CBL) and CBL-interacting protein kinase (CIPK) [38,39]. The activity and specificity of NRT1.1 are regulated by two CIPKs, CIPK8 and CIPK23, which interact with NRT1.1 and mediate low- and high-affinity responses, respectively [40]. At high NO_3_^−^ concentrations, NRT1.1 activates the expression of Arabidopsis nitrate regulated 1 (ANR1), a transcription factor that promotes lateral root growth and proliferation [41]. At low NO_3_^−^ concentrations, NRT1.1 inhibits lateral root development by controlling auxin levels and meristem activation [42]. To transmit the NO_3_^−^ signals from NRT1.1 to the nucleus, calcium acts as a secondary messenger that modulates the expression of NO_3_^−^-responsive genes. Three calcium-dependent protein kinases (CPK10, CPK30, and CPK32) and their partner CBLs are involved in this process [43].

Several NRTs are involved in the loading and unloading of NO_3_^−^ from the xylem and phloem, which affects the distribution and availability of NO_3_^–^ within the plant. For example, NRT1.5, NRT1.8, and NRT1.9 are expressed in the xylem and phloem, and they function in the influx/efflux, removal, and loading of NO_3_^−^ from/to the root vascular tissues, respectively [43,44]. Other studies have shown that overexpressing NRTs in different plant species can improve N-use efficiency (NUE) as well as plant biomass to nitrogen input. However, the effects of NRT overexpression may depend on the tissue specificity, NO_3_^–^ concentration, and interaction with other genes. For instance, only NRT1 transporters have been successfully overexpressed in both leaves and roots to enhance NUE, while the role of NRT2 transporters in this process is still unclear [45]. Recently, a chimeric NRT1 transporter, AtNC4N, was overexpressed in the phloem of old leaves in Arabidopsis, rice, and tobacco, and it increased NO_3_^–^ uptake and NUE under low nitrate conditions [46]. Another NRT1 transporter, OsNRT1.1A (OsNPF6.3), was found to improve NUE, flowering time, yield, and maturity in rice [47]. Several other NRTs, such as OsNRT1.1B (OsNPF6.3B), OsNRT2.1, OsNRT2.3a/b, OsPTR9, OsAMT1.1, and qNGR9, have also been reported to enhance NUE under high or low nitrate levels in rice [47,48,49,50].

Moreover, some genes that regulate or interact with NRTs have been identified as potential targets for improving NUE in plants. For example, the transcription factor OsNAC42 activates the expression of several NRTs and increases nitrate assimilation and NUE in rice. The nitrate reductase gene OsNR2 converts nitrate to ammonium, affecting the expression of NRTs and other nitrogen-related genes in rice. The nitrate transporter OsNPF4.5 is regulated by a microRNA (miR827) and modulates nitrate distribution and remobilization in rice. These studies demonstrate the complexity and diversity of nitrate transport and signaling in plants as well as provide valuable insights for improving NUE and crop production [51]. Notably, NUE can be improved by transcription factors (TFs) in multiple crops [52,53]. TFs are primarily proteins (rarely RNA) that control the transcription of genes by interacting with their unique promotor sequences [54]. Numerous TF families have been found and proven to interact with particular DNA sequences on regulatory regions of NO_3_^−^-responsive genes. According to reports, these TF families act as essential regulators of the plant N response, which encompasses MADS-box [55], NLP [53], b-ZIP [56], LBD [57], NAC [58], MYB [59], GARP [60], TCP [61], AP2/ERF [62], and zinc/finger proteins [63].

NRT1, NRT2, and NRT3 have different structural conformations (Figure 1). As a result, the Pfam database collects these genes under distinct families: PF00854 for NRT1, PF07690 for NRT2, and PF16974 for NRT3 [64]. PF00854 refers to the proton-dependent oligopeptide transporter (POT) family (also known as the peptide transporter (PTR) family) and is a class of energy-dependent transporters discovered in organisms ranging from bacteria to humans [65]. However, some family members are NO_3_^–^ permeases, while others are in histidine transport [66]. The NRT1 transporter adopts a standard major facilitator superfamily (MFS) fold, which is described by 12 transmembrane helices (TMHs) with a pseudo 2-fold axis connecting the N-terminal (TMH1–6) and C-terminal (TMH7–12) domains [67].

NRT2 members belong to the PF07690 (MSF) family, representing the largest family of secondary transporters, with members ranging from *Archaea* to *Homo sapiens*. MFS proteins target a diverse range of substrates in both directions across the membrane, including ions, carbohydrates, lipids, amino acids and peptides, nucleosides, and other small molecules, in many cases catalyzing active transport by converting the energy stored in a proton electrochemical gradient into a concentration gradient of the substrate [68]. NRT3 candidates lie under the PF16974 family, which contains a C-terminal transmembrane domain required for high-affinity nitrate absorption [69,70]. NRT3 is implicated in the inhibition of lateral root initiation in the presence of high sucrose-to-nitrogen ratios in the medium. As a result, the NRT2 and NRT3 component systems are believed to be implicated in the signaling pathway that integrates nutritional inputs for the regulation of lateral root architecture [71]. In *Arabidopsis*, the functional unit of the high-affinity nitrate influx complex is most likely a tetramer composed of two NRT2 and NRT3 subunits each [72].

### 3.2. Plant Ammonium Transporters

As mentioned earlier, NO_3_^−^ and NH_4_^+^ are the principal inorganic nitrogen sources absorbed by plant roots [73,74]. Because NH_4_^+^ requires less energy than NO_3_^−^ assimilation, NH_4_^+^ is the preferred form of N uptake when plants grow under N deficiency [75,76]. Although NH_4_^+^ uptake is more efficient, high absorbed amounts of NH_4_^+^ by the plant can be hazardous [77,78,79]; thus, the NH_4_^+^ uptake system in plants receives special attention [80,81,82]. Recent research indicates that AMTs also play a role in a variety of other physiological processes such as transporting NH_4_^+^ from symbiotic fungi to plants [83,84], delivering NH_4_^+^ from roots to shoots [85], transmitting NH_4_^+^ in leaves and reproductive organs [86,87,88], and encouraging resistance to plant diseases through NH_4_^+^ transport [89,90]. Aside from being a transporter, many AMTs are essential for root growth in the presence of NH_4_^+^ [81,91,92,93]. To avoid the adverse effects of insufficient or excessive NH_4_^+^ intake on plant growth and development, AMTs activities are fine-tuned not only at the transcriptional level through the involvement of at least four transcription factors [94,95,96,97] but also at the protein level through phosphorylation [98,99], pH [100], and endocytosis [101].

The NH_4_^+^ transporter domain (PF00909) comprises two structural copies of five helices plus one extra C-terminal helix. It has been defined as an 11-times-spanning membrane channel. Ideal NH_4_^+^ transporter domain is composed of 11–12 transmembrane spots, with feature sequences “D (F Y W S) A G (G S C) X2 (L I V) (E H) X2 (G A S) (G A) X2 (G A S) (L F)” at transmembrane spot 5 and “D D X (L I V M F C) (E D G A) (L I V AC) X3 H (G A L I V) X2 (G S) X (L I V A W) G” at transmembrane region 10 [81]. AMTs can be divided into two subtypes: AMT1 and AMT2 or methylammonium permeases (MEP) [81,85,102]. Figure 1 illustrates the diversity and complexity of the N transporters in plants and their distinctive features.

### 3.3. Plant Nitrite Transporters

Cytosol is the first cellular compartment to encounter NO_3_^−^ in the root that needs to be converted to NO_2_^−^ to make organic compounds such as amino acids. The cytosolic NO_2_^−^ is toxic to the cytosol and needs to be transported to the chloroplasts, where it is reduced to NH_3_ by NIR [103]. Cytosolic NO_2_^−^ needs to be actively transported across the membrane of the chloroplasts by a specific transporter to avoid accumulation and toxicity. Several types of plastids and plasma membrane-localized NIR have been described in *Escherichia coli*, *Cyanobacteria*, green algae, red algae, *Aspergillus nidulans*, diatoms, and a haptophyte, but there was no evidence of a nitrite-specific plastid and plasma membrane transporter in higher plants [103,104]. Consequently, it has been hypothesized that NO_3_^−^ and NO_2_^−^ enter the plasma membrane via similar transporters due to previously observed competitive reduction of NO_2_^−^ absorption by NO_3_^−^ and vice versa [105].

In 1988, Brunswick and Cresswell found that illuminated chloroplasts take up NO_2_^−^ in vitro [104]. In 2007, Saguira et al. revealed that CsNitr1-L is the NO_2_^−^ transporter in the chloroplasts of plants and belongs to the proton-dependent oligopeptide transporter (POT) family [106]. CsNitr1-L, a homologue of NRT1 transporter, can transport NO_2_^−^ depending on the energy availability. In 2013, Kotur et al. used ^13^NO_2_^−^ to characterize nitrite influx and nitrite-specific transporter into roots of *Arabidopsis thaliana* [105]. Fortunately, they successfully confirmed that Arabidopsis roots include a transporter specific for nitrite [105]. Furthermore, in 2014, Maeda et al. succeeded in characterizing two genes, AT5G62720 (AtNITR2;1) and AT3G47980 (AtNITR2;2), which encode an integral membrane protein related to the NO_2_^−^ transport activity from Arabidopsis, and these two genes were detected based on homologs genes from cyanobacterial genomes [107].

AtNITR2;1 and AtNITR2;2 are HPP-domain-containing proteins (Figure 1). These two transporters belong to PF04982, constituting integral membrane proteins with four transmembrane-spanning helices [108]. Interestingly, their work has great value, as the HPP proteins had no functional annotation before their work [107]. Moreover, the kinetic analyses showed that the proteins that encode from these previous two genes of Arabidopsis have a much higher affinity for NO_2_^−^ than the cyanobacterial proteins [107]. AtNITR2;1, isolated from the chloroplasts mutant lines, showed much lower NO_2_^−^ uptake activity than the chloroplasts isolated from the wild-type Col-0 plants. AtNITR2;2, expressed in roots and not detectable in shoots, had a putative transit peptide similar to that of AtNITR2;1 and demonstrated low but significant activity of NO_2_^−^ transport in the cyanobacterial cell. Finally, it was discovered that NO_2_^−^ increased the expression of AtNITR2;1 and AtNITR2;2 under the direction of NIN-like proteins, indicating that these two proteins are nitrate-inducible parts of the nitrite transport system of plastids.

## 4. Nitrogen Metabolism

N is among the most widely distributed elements in the lithosphere, atmosphere, hydrosphere, and biosphere [109]. The lithosphere contains 94% of all nitrogen on Earth (e.g., NO_3_^−^, NO_2_^−^, and NH_4_^+^), with the remaining 6% in the atmosphere (e.g., NO, N_2_O, and N_2_) and a trace (0.006%) in the hydrosphere and biosphere (e.g., NO_3_^−^, NO_2_^−^, and NH_4_^+^). N is the fourth most prevalent element in the biosphere after oxygen, carbon, and hydrogen and is an essential component of total biomass [110,111]. The relative quantity of N in the biosphere reflects the relevance of N to living organisms. Although organic N, such as amino acids, hormones, and nucleic acids, is vital to life on Earth [112,113,114,115], The global N cycle is dominated by inorganic N molecules (i.e., NO_3_^−^, NO_2_, nitrous oxide (N_2_O), NO, dinitrogen (N_2_), and NH_4_^+^) [112,116,117,118]. Figure 2 summarizes all N metabolism pathways.

NO_3_^−^ is the most oxidized form of N, while NH_4_^+^ is the most reduced form. Many microorganisms, mainly microbes, drive the global N cycle by metabolizing N through a variety of redox processes for energy transduction, detoxification, or assimilation via various sub-pathways such as N fixation (KEGG;M00175), assimilatory NO_3_^−^ reduction (KEGG;M00531), dissimilatory NO_3_^−^ reduction (KEGG;M00530), denitrification (KEGG;M00529), nitrification (KEGG;M00528), complete nitrification, complete ammonia oxidation (comammox) (KEGG;M00804), and anammox (KEGG;M00973).

Moreover, each sub-pathway from these previous sub-pathways consists of a list of genes and their encoded enzymes that are involved in the various biogenic processes in the nitrogen cycle. The nitrogen fixation path consists of nitrogenase molybdenum-iron protein alpha chain (NIFD), vanadium-dependent nitrogenase alpha chain (VNFD), and nitrogenase delta subunit (ANFG) genes [119,120,121].

In this sub-pathway, these previous genes were used to convert N_2_ to NH_3_ by oxidoreductase reaction. NH_3_ is widely used in different vital pathways for living organisms, such as arginine biosynthesis (KEGG;map00220), purine metabolism (KEGG;map00230), alanine, aspartate, and glutamate metabolism (KEGG;map00250). Also, NH_3_ is involved in glycine, serine, and threonine metabolism (KEGG;map00260) as well as cyanoamino acid metabolism (KEGG;map00460), glyoxylate and dicarboxylate metabolism (KEGG;map00630), and lipoic acid metabolism (KEGG;map00785). NH_3_ is a component in biosynthesis of secondary metabolites (KEGG;map01110), microbial metabolism in diverse environments (KEGG;map01120), carbon metabolism (KEGG;map01200), biosynthesis of amino acids (KEGG;map01230), proximal tubule bicarbonate reclamation (KEGG;map04964), collecting duct acid secretion (KEGG;map04966), protein digestion and absorption (KEGG;map04974), *Vibrio cholerae* infection (KEGG;map05110), and epithelial cell signaling in *Helicobacter pylori* infection (KEGG;map05120) [119,120,121].

In addition, the anammox path consists of hydrazine dehydrogenase (HDH), hydrazine synthase subunit (K20932), and nitrite reductase (NO forming) (NIRK) genes, and these previous genes can convert NO_2_ to N_2_ or NH_3_ using nitrite reductase enzyme [122,123]. In that context, the denitrification path consists of four genes: nitrous-oxide reductase (NOSZ), NIRK, nitrate reductase (cytochrome) (NAPA), and nitric oxide reductase subunit B (NORB) [124,125,126]. In this sub-pathway, the denitrification enzymes convert NO_3_^−^ to N_2_ through NO_2_, NO, and N_2_O. Furthermore, assimilatory and dissimilatory NO_3_^−^ reduction paths have different genes, such as nitrate reductase/nitrite oxidoreductase, alpha subunit (NARG, NARZ, and NXRA), NAPA, nitrite reductase (NADH) large subunit (NIRB), nitrite reductase (cytochrome C-552) (NRFA), ferredoxin-nitrate reductase (NARB), nitrate reductase (NAD-(P)-H) (NR), assimilatory nitrate reductase catalytic subunit (NASC and NASA), nitrite reductase (NAD-(P)-H) (NIT-6), ferredoxin-nitrite reductase (NIRA), and nitrite reductase (NAD(P)H) large subunit (NASD and NASB) that can convert NO_2_ to NH_3_ [127,128,129,130]. Figure 2 and Table 1 show the enzymes and genes involved in the N cycle.

### 4.1. GS and GOGAT

N uptake for plant growth has been extensively studied and discussed previously [172]. N uptake involves an active transport across the plasma membrane of both root epidermal and cortical cells. The gene expression patterns, post-translational regulation, and the localization of NRTs and AMTs enable directed transport from the epidermis to vascular tissue and acclimation to various concentrations of N in the soil [173]. Mostly, the plant uptakes N from the soil in the form of NO_3_^−^ that is reduced by NR to NO_2_^−^, which is then imported into the chloroplast and reduced further by NIR into NH_4_^+^. NO_2_^−^ reduction to NH_4_^+^ occurs in the stroma of chloroplasts and is catalyzed by the enzyme NIR. NIR reduces NO_3_^−^ to NO_2_^−^ using electrons from NAD(P)H, and the reduced ferredoxin (Fdred) generated due to the photosynthetic electron transport is the electron donor for NO_2_^−^ reduction. NH_4_^+^ is combined with glutamate to be assimilated into glutamine [174]. NR is regulated by NO_3_^−^, N, light, growth conditions, hormones, N metabolites, and phosphorylation [175,176,177,178].

In various biological systems, the interconversion of glutamate and glutamine plays a central role in N assimilation, transport, and recycling. GS and GOGAT are vital enzymes orchestrating these interconversions, ensuring optimal N utilization and maintaining N homeostasis [112]. The roots are the principal site of NH_4_^+^ assimilation, where the enzyme GS facilitates this conversion [179,180]. Following GS action, one of the amino groups from glutamine is transferred to 2-oxoglutarate (2-OG) to synthesize glutamate, and this reaction is catalyzed by GOGAT (Figure 3). The coordinated action of GS and GOGAT is responsible for these interconnected processes [181,182,183]. Experimental studies involving reverse genetic analysis have demonstrated that the absence of genes encoding NH_4_^+^-responsive GS/GOGAT isoenzymes reduces NH_4_^+^ assimilation, particularly in the roots, resulting in impaired growth [184,185,186,187]. These findings suggest that the NH_4_^+^-responsive forms of GS/GOGAT have a central role in the primary assimilation of NH_4_^+^ within the roots [188]. As the expression of these isoenzyme-encoding genes increases with NH_4_^+^ availability, it is plausible to infer that plants possess a regulatory transcriptional network that modulates their gene expression in response to NH_4_^+^ availability [185].

Plant GS (EC6.3.1.2), a highly conserved enzyme, catalyzes glutamine synthesis from glutamate and ammonium ions, conserving the N atom in an organic molecule. This conversion is fueled by the energy released through the hydrolysis of ATP to ADP, making GS a crucial component of N metabolism across different life forms [189]. In addition, the GOGAT in synthesizing glutamate involves transferring the amide amino group of glutamine to 2-OG through a reductant-driven process, yielding two glutamate molecules [190]. Within the realm of plants, this enzyme exists in two distinct variants: one produces the ferredoxin (Fd) as the electron donor (EC 1.4.7.1), while the other employs NADH as the electron donor (EC 1.4.1.14). The Fd-dependent form of the enzyme is notably abundant within the chloroplasts of photosynthetic tissues. In these regions, it utilizes light energy directly to serve as a reductant source. Figure 3 provides an overview of the N cycle, transport, and utilization in plants.

Conversely, the NADH-dependent variant, also present within plastids, is primarily concentrated in non-photosynthesizing cells. The reductant is sourced from the pentose phosphate pathway [191]. Notably, the expression of the Fd- and NADH-dependent forms of glutamate synthase seems to exhibit dissimilar patterns across distinct plant tissues. It has been suggested that in most plants, two genes encode each form of GOGAT [190].

### 4.2. Isoforms of GS and GOGAT

According to subcellular distribution patterns, investigations into plant GS enzymes have identified two main categories. The initial group, designated as GS1, is primarily situated within the confines of the cytoplasm. Conversely, the second group, GS2, is predominantly found within plastid structures [192,193]. Various multigenic families are responsible for encoding multiple isoforms of GS1, whereas the plastidial GS2 arises from a limited number of nuclear genes [194]. Typically, GS2 is linked to the process of (re)assimilation of NH_4_^+^ in leaves, whereas GS1 is associated with recycling N within the plant. However, the relative activity levels exhibited by GS1 and GS2 are subject to variation based on species, specific plant organs, N sources, developmental stages, and prevailing environmental conditions, including abiotic stress factors. This variability underscores the intricate involvement of these isozymes, reflecting a multifaceted role [193,195]. In wheat research, a comprehensive exploration has unveiled four distinct variations of GS genes: GS1, GS2, GSr, and GSe [192]. Employing advanced high-performance liquid chromatography (HPLC) methods, ref. [196] successfully separated GS and discovered two subtypes in leaves: cytoplasmic subtype GS1 and chloroplast subtype GS2. Both subtypes’ content and activity fluctuate in tandem with developmental processes. The subcellular localizations of GSr and GSe align with the cytoplasmic compartment [197]. In addition, the advancement of genome sequencing has unveiled various isoenzymes of GS/GOGAT present in plants. Among these, specific isoenzymes such as GS1,2 and NADH-GOGAT1 from rice [198] as well as GLN1,2 and NADH-GOGAT (GLT1) from *Arabidopsis* [185,199] are expressed in plant roots in response to NH_4_^+^ supply. These isoenzymes’ expression at transcript and protein levels highlights their specificity in NH_4_^+^ assimilation.

### 4.3. Nitrogen Metabolism in Microalgae

Microalgae are unicellular aquatic microorganisms with sizes between 1 µm and 2 mm. They can generate biomass using light and carbon dioxide through the photosynthetic process. Due to their remarkable tolerance for environmental stress, they can be grown in various environmental conditions, including freshwater, saltwater, or wastewater [200]. After C, N is quantitatively the most significant element contributing to the dry matter of microalgal cells, accounting for between 1 and 10% of their dry weight. C and N metabolism are related in microalgae since they share the following: (a) C provided directly by respiration of CO_2_ or assimilated organic C and (b) the energy obtained by the TCA cycle and mitochondrial electron transport chain. To create the amino acids glutamate, glutamine, and aspartate, the primary absorption of inorganic N (NH_4_^+^) needs carbon skeletons in the form of keto acids (2-OG and oxaloacetate) as well as energy in the form of ATP and NADPH [201]. There are four fundamental phases in the NO_3_^−^ assimilation mechanism in eukaryotic microalgae and plant leaves: NAD (P) H-nitrate reductase catalyzes NO_3_^−^ reduction into NO_2_^−^ in the cytosol of cells, which is required for (a) NO_3_^−^ transport into the green cell, and (b) NO_3_^−^ reduction into NO_2_^−^ in the cytosol of cells. In addition, in Cyanobacteria, ferredoxin-nitrate reductase catalyzes the reduction of NO_3_^−^ to NO_2_^−^, (c) NO_2_^−^ transport into the chloroplast of cells via NAR1 family and its subsequent reduction into NH_4_^+^ in a six-electron reaction catalyzed by ferredoxin-nitrite reductase, and (d) the process mediated by the ATP-dependent GS- GOGAT cycle, which results in the formation of L-glutamate by the incorporation of NH_4_^+^ into the carbon skeleton of 2-OG [103]. Microalgae mostly use the GS-GOGAT pathway to assimilate NH_4_^+^ as well as via the amination of 2-OG catalyzed by NAD(P)H glutamate dehydrogenase [NAD(P)H-GDH [202,203].

## 5. The Role of Nitrogen in Plant Response to Abiotic Stress

Plants must endure abiotic stressors, including drought, salt, and severe temperatures, because they are immobile. The spread of plants is significantly constrained by these stresses, which also change plant growth and development and lower agricultural output [204]. Since N is considered the most crucial nutrient for plant growth quantitatively, plants have developed effective methods to control N levels in response to complicated stresses [14].

### 5.1. Nitrogen and Drought Stress

Drought stress significantly damages crops worldwide, particularly in arid and semi-arid regions [205]. Hence, plants have evolved to adapt their resilience against drought via avoidance (stomata adjustment to reduce transpiration rate) and tolerance (antioxidants and osmolytes production) of drought [206] (Figure 4). Remarkably, the concentrations of NO_3_^−^ and NH_4_^+^ have impacts on plants’ underwater limitations. The application of NH_4_^+^ alleviates the effect of drought on plant development, while NO_3_^−^ has the opposite impact [54]. In that sense, N fertilizers might enhance the physiological responses of plants under drought stress by elevating the concentrations of N and chlorophyll and promoting PSII photochemical activity [207]. It has been shown that adding N (NH_4_NO_3_) improved water-use efficiency (WUE) by increasing plant dry mass and decreasing water loss [206].

Additionally, N may lessen drought’s inhibitory effects on photosynthesis and prevent C starvation [208]. On the other hand, low N availability can make plants more sensitive to dehydration and cause protein degradation, which lowers the amount of N-containing osmolytes like proline [206,208]. N supply enhances osmotic adjustment and the activity of antioxidant enzymes via proline accumulation induction to mitigate the physiological harm from drought stress. However, the N effects depend on the ionic form of N supplied [209]. NH_4_^+^ is essential in promoting plant drought tolerance by increasing root numbers (surface area) and subsequently improving and facilitating water uptake. Also, decreased aerenchyma development may contribute to NH_4_^+^-promoted drought tolerance [14]. The influence of NO_3_^−^ on drought responses is linked to NO_3_^−^ transport/assimilation in plants. Numerous NO_3_^−^ transport/assimilation-related genes, such as NRT2.5, GS, and GOGAT, are repressed in response to drought stress [210]. Interestingly, a high N concentration enhances root growth and photosynthetic traits, alleviates drought stress via stomatal conductance, and increases the photosynthetic rate [211]. Plant nutrient management is crucial to promote abiotic stress tolerance in cotton plants [212]. The use of nitrogen fertilizer either in the soil or foliar spray can decrease the toxicity of reactive oxygen species (ROS) by increasing the peroxidase (POD), catalase (CAT), and superoxide dismutase (SOD) levels in plant cells [205,212]. These enzymes improve the photosynthetic rate, promoting stress tolerance by scavenging ROS and reducing photooxidation to preserve chloroplast membrane integrity [212,213,214]. Applying nitrogen fertilizers (N60 kg.ha^−1^) can increase the tolerance of drought stress in wheat plants combined with the highest yield [205].

Under drought stress, GS1 and GS2 exhibited differential expression among different plant organs and species, with a prominent increase of GS1 in roots and leaves, while the total GS activity showed a decrease in potato root [215,216]. A comparative proteomic analysis under drought stress conditions in wheat was studied [217]. The researchers found that drought-tolerant wheat varieties exhibited a significant upregulation of GS2 expression, indicating its potential role in enhancing drought tolerance. Similar results were obtained by Nagy et al. [218], who explored GS activity in wheat leaves and observed that older leaves had lower GS activity than flag leaves in drought-tolerant and sensitive varieties under normal conditions. During drought stress, GS activity in sensitive varieties’ flag leaves decreased compared to tolerant varieties. This suggested that GS activity could indicate drought stress sensitivity and tolerance, with elevated cytoplasmic-to-plastid GS ratio and premature senescence as markers. Exposure to heavy metals such as cadmium, molybdenum, copper, and aluminum led to the down-regulation of GS and decreased its total activity in different plant species, including rice, soybean, tomato, and tea [219,220,221,222].

Like all organisms, plants must respond to various extreme environmental cues by responding to various internal signals. NO is a small lipophilic molecule that diffuses through plant cell membranes as active signals to thrive and survive under those conditions. In addition, NO is a relevant signal molecule in many plant processes. Several pathways for NO production, either oxidative in the presence of O_2_ or reduction from NO_2_^−^, have been described in plants [223]. NO regulates gene expression, modulates enzyme activities, and acts as a metabolic intermediate in energy regeneration. However, the mechanisms and proteins responsible for its synthesis remain complex, with many unresolved questions.

Moreover, several studies showed the involvement of NO in reproductive processes, control of development, and the regulation of physiological responses such as stomatal closure. NO also regulates the expression of several genes involved in the synthesis of and response to pathogen attacks as well as reproductive mechanisms that operate during pollen recognition by the stigma [224,225,226,227]. NO is a high redox to reactive N species (RNS) formed in living cells under normal and stressful biotic and abiotic conditions, and NO also functions as a detoxifier and lessens adverse effects when a plant’s ROS content reaches dangerous levels [228]. It was reported that the synthesis of NO increased in plant cells under drought stress conditions, and it has been suggested that this may help enhance plants’ acclimation responses to cope with water limitation [229]. Drought stress enhanced NO synthesis in beat, pea, tobacco, cucumber, grapevine, and rice plants; likewise, the activity of NO synthase (NOS)-like and NO release rate increased under dehydration conditions [230]. NO has a vital role in respiratory function, called pathways of electron transport in mitochondria; it adjusts ROS as well as activates defense strategies via promoted antioxidant formation in plants under various abiotic stresses [228]. NO plays a crucial role in mediating hormonal activities and influencing gene expression as well as protein activity during sensing and signaling processes that enable plants to survive under environmental stressors. The exogenous supply of NO increased drought tolerance in various plants such as maize, rice, and cucumber due to the increase of antioxidant defense, which acts to scavenge ROS, improve cellular membrane stability, and preserve photosynthesis as well as water status [230,231].

Drought induces the synthesis of abscisic acid (ABA), which regulates several critical plant processes like seed germination and stomatal conductance, with the help of signaling molecules like NO and ROS [232]. Interestingly, NO acts as a critical signaling molecule promoting ABA-induced stomatal closure in case of hyperosmotic stress [233]. The plants’ main enzyme responsible for NO production is the molybdoenzyme NR, which contains two subunits, each bearing three prosthetic groups: FAD, heme b557, and molybdenum. In an NR subunit, molybdenum is bound to a tricyclic pyranopterin and chelated by a dithiolene named the molybdenum cofactor (Moco). Amongst the reducing sources of NO in plants, NO_2_^−^ can be reduced to NO by a family of five molybdenum (Mo)-containing enzymes, which include NR, xanthine oxidase reductase (XOR), aldehyde oxidase (AO), sulfite oxidase (SO), and mARC [234,235].

In a study by Berger et al. (2020) three NR genes (MtNR1, MtNR2, and MtNR3) were discovered in the genome of *Medicago truncatula*, and they study addressed their expression, activity, and potential involvement in NO production during the symbiosis between *M. truncatula* and *Sinorhizobium meliloti* [235]. These results revealed that MtNR1 and MtNR2 gene expression and activity are correlated with NO production throughout the symbiotic process and that MtNR1 is mainly involved in NO production in mature nodules [235]. However, when plants were grown in soil containing NH_4_^+^, the functional role of NR was eliminated [236]. Moreover, NO can alleviate the negative impact of ROS through lipid peroxidation, enhancing photosynthesis as well as the expression of antioxidant enzymes via mitogen-activated protein kinase (MAPK) and other signaling pathways [228]. It was found that exogenous application of NO caused a significant increase in seedling growth of cucumber by decreasing seedlings injury and enhancing the antioxidant activities and chlorophyll and free proline contents under stress conditions [237]. Moreover, an exogenous supply of NO in wheat markedly preserved high relative water content via decreasing transpiration and increasing K content and mitigated cell membrane damage [238]. Figure 4 explains how NO can work as a signaling molecule that can protect plants from drought stress by modulating various physiological and biochemical processes.

### 5.2. Nitrogen and Salinity Stress

Salinity stress has a destructive impact on the metabolism and productivity of crops worldwide, resulting in the remarkable conversion of agricultural arable soil into unproductive land [239]. The high level of salt ions in agricultural soil may be harmful to plants, causing salt stress, where sodium (Na^+^) and chlorine (Cl^−^) concentrations increase in saline soil and affect the natural physiological process in plants [240]. N application is an efficient way to improve plants under saline conditions. N is a necessary nutrient that mediates various molecular, physiological, and cellular responses essential to plant survival as well as different signal transduction pathways linked to plant defense systems and salt-stress tolerance [241,242]. Moreover, salt stress affects ammonification and nitrification in the soil because Cl^−^ competes with NO_3_^−^, and NH_4_^+^ competes with Na^+^ and creates ion toxicities and inequalities, which can limit the processes of N uptake, transport, and assimilation [243,244]. Salt stress has inhibited various enzyme activities that contribute to N assimilation in plants such as maize, tomato, rice, cowpea, and mung bean, reducing N uptake and utilization [245].

Interestingly, it was reported that the interaction between salt stress and N had an impact on plant growth, and the effect of N on salt-exposed plants was dependent on the N rate [240]. In the case of maize, excessive N treatment can successfully mitigate the deleterious effects of salt stress [246]. Similar findings in tomatoes showed that supra-optimum N was better than optimum N in reducing salt stress [247]. Under salinity stress, the application of N significantly increased both N and K uptake and decreased Na accumulation in wheat seedlings. Furthermore, N application can protect wheat seedlings in saline conditions via the upregulation of osmolytes, antioxidant system, and secondary metabolite accumulation [239]. Even though excessive N application exacerbated the negative effect of salt stress on wheat and rice, moderate N supply could ameliorate the deleterious effect of salt stress. Therefore, applying an appropriate N rate is the key to mitigating the adverse effects caused by salt stress [240].

On the contrary, it was reported that the application of NO_3_^−^ resulted in an increase of Na^+^ uptake and loading of the sodium into the xylem, increasing root inhibition caused by the salinity. Na^+^ and NO_3_^−^ co-transportation was suggested based on NO_3_^−^-dependent Na^+^ uptake data at various Na^+^ concentrations [14]. Similarly, NO_3_^−^-dependent transport mechanisms in salt-stress environments encourage Na^+^ ion uptake and loading into the xylem, which may represent a main pathway for Na^+^ accumulation in Arabidopsis shoots [241].

It is worth noting here that GS isoforms imply a multifaceted role for GS in navigating through various dimensions of abiotic stress tolerance [193,248]. For instance, in rice plants exposed to salinity stress, GS1 increased in the root at the seedling stage and old leaves with no change in young leaves, while GS2 was down-regulated in old and young leaves [249].

It has been demonstrated that NO can act as a signal to induce the resistance of salt stress by raising the K:Na ratio in plant cells, which is based on an increase in the H1-ATPase activity of plasma membrane in calluses from maize seedlings, reed (*Phragmites communis*) plants, and poplar trees [230]. Exogenous NO can improve salt tolerance under salinity conditions by stimulating the K^+^/Na^+^ ratio, the Na^+^/H^+^ antiport in the tonoplast, and proton pump activity [250]. Under salt stress, NO improves the absorption and translocation of several macro- and micronutrients, including K, Fe, Mg, and Zn, which improve respiration and chlorophyll biosynthesis [251]. The uptake of mineral nutritional elements was improved due to NO application, thereby diminishing the harmful effect of high salinity on strawberry plants [252]. The exogenous NO supply can protect chickpea plants from salt stress-induced oxidative harm by promoting the biosynthesis of photosynthetic pigment and antioxidant enzymes osmolyte accumulation, thereby improving plant development under saline stress [253]. Furthermore, under salt stress conditions, exogenous NO treatment enhanced wheat seed germination by converting starch to sugars, where the application of NO led to an elevation in the effectiveness of α- and β-amylase enzymes [251,254]. Under salt stress, NO treatment boosted plant development and demonstrated a protective ability against salinity-triggered oxidative stress by increasing the effectiveness of POD, SOD, and CAT enzymes in *Triticum aestivum* [252].

## 6. Hazardous Effects of N Fertilizers

Nutrient deficiency and/or availability are crucial to plant production [255,256]. One of the severe issues the world faces is the adverse effects of fertilizers on the environment. Farmers have been using fertilizers since ancient times, but long-term use has affected soil fertility adversely. Due to over-fertilization in many agricultural areas, environmental pollution was established [257]. N pollution in groundwater is a severe problem worldwide, affecting animal and human health; in addition, NO_3_^−^ leaching decreases N availability for crops and can cause water pollution [258]. The improper application of fertilizers damages the farmlands, leading to heavy metal contamination and soil erosion. While ammonium sulfate causes soil pollution, nitrogen oxides and NH_4_^+^ cause air pollution, and NO_3_^−^ is the final breakdown product accumulating in groundwater [259,260]. Since NO_3_^−^ levels above 10 mg L^−1^ N cause oxygen debt in the blood, the U.S. Environmental Protection Agency set human drinking water to maintain nitrate levels below a maximum contaminant level (MCL) at a critical value of 10 mg L^−1^ N [261]. In Argentina, the national food law set the limit for human consumption at 11.3 mg L^−1^ N [258]. Consumption of NO_3_^−^-rich water has been associated with methemoglobinemia (blue child syndrome) and different forms of cancer [262,263,264]. Furthermore, diabetes, adverse reproductive outcomes (especially neural tube defects), and thyroid conditions are related to NO_3_^−^-contaminated drinking water.

A notable study by Maghanga et al. (2013) was conducted to monitor changes in surface water NO_3_^−^ levels in ten rivers within a Kenyan tea plantation for three years. Water samples were obtained before and after fertilizer application in 2004, 2005, and 2006. For three years, there was no established trend between surface water nitrate levels and the time of fertilizer applications. However, the highest nitrate–nitrogen levels were in thr river Temochewa during the first fertilizer applications (4.9 mg L^−1^ to 8.2 mg L^−1^). Furthermore, fertilizer application increased NO_3_^−^ levels, and the study indicated that initial nitrate–nitrogen levels in most rivers were high, causing surface water contamination [261].

## 7. Promoting N-Fixation through Diazotrophic Microbiota

Rapid human population growth and declining agricultural soils intensify resource competition, affecting human food hunger and sustainable agriculture systems worldwide [5]. Hence, developing alternative strategies to increase the current food demand is an inevitable need [265]. Accordingly, crop productivity is the pillar of nutritional food security and heavily depends on applying N fertilizers [266]. Applying N fertilizers consumes vast amounts of energy and causes excessive harm to the environment and health, expensive costs, soil fertility reduction, and other negative consequences [267,268]. Therefore, increasing plant NUE is essential for developing sustainable agriculture [269]. Ultimately, N fixation has multiple benefits through improving the yields of landraces that often receive little attention from breeders and empowering farmers using a sustainable and less expensive strategy. Additionally, it reduces the need for fossil fuels requiring synthetic N fertilizers [270,271]. Overall, the symbiotic relationship between leguminous plants and *Rhizobia* is a powerful adaptation strategy that enables these plants to reduce their dependence on synthetic fertilizers and thrive in N-deficient soils. Improving the microbial activities that boost plant growth is necessary to increase food production [272]. In that sense, several beneficial mutualistic microbes have been discovered; however, their dependable utilization as biofertilizers in soil conditions is still challenged. Thus, improving the microbial inoculants and exploring their diversity performance in the external habitat could lead to understanding the knowledge gap between the microbial performance in vitro and the external habitat [273].

Bacterial endophytes associated with crops, i.e., corn and wheat, can produce various sugar compositions, such as arabinose [274], to stimulate the production of a thick gelatinous layer (mucilage) around the root systems of these crops. This layer allows crops to better uptake fertilizer by forming a continuous boundary layer between roots and soil particles, protecting the root system from drought and supporting other microbes (Figure 5). Most interestingly, mucilage structures associated with complex microbiota contribute to nutrient acquisition and play a vital role in plant growth and defense [271].

Improving the sustainability of biological nitrogen fixation (BNF) to cereal crops is a future reason to enhance crop productivity, representing a significant breakthrough in N-fixation research. To achieve this goal, engineering key regulators of BNF by activating the symbiotic signaling pathway between diazotrophic bacteria and cereal crops is a promising strategy for sustainable development.

Fortunately, establishing new omics techniques such as metabolomics, metagenomics, metatranscriptomics, and metaproteomics will help in-depth research and identify the rhizospheric microbes using next-generation sequencing approaches [275]. Accordingly, manipulating the promising microbes that replace synthetic fertilizers is a solution. *Rhizobium*–plant symbiosis is now becoming challenging with diverse genetic modifiers of the symbiosis relationship not only in the genomic variability of partners but also in the soil microbiota. In that sense, there is a future direction to improve the specificity of the partnership during symbiotic colonization using different methods (e.g., axenic cultures) by eliminating other microbes. Therefore, identifying and developing bioinoculant strains and genotypes by adapting and evolving elite strains in both lab-scale analyses and trials to open-field applications to increase crop yield becomes crucial. Figure 5 shows how microbes can help plants increase their N availability by colonizing their roots, releasing exudates, and enhancing nutrient uptake.

## 8. Microbial Alternatives to Synthetic Fertilizers

Fertile soil is a primary component in the backbone of sustainable agriculture; however, land degradation and rapid desertification cause an estimated global loss of 24 billion tons of fertile arable land [276]. Noticeably, eutrophication and decline of soil fertility result from massive application of chemical fertilizers, causing significant environmental concern [277]. Moreover, with the limited N resources and increasing agricultural demands for N supplies, there is a need to discover diverse mutualistic interactions between plant roots and rhizosphere microbiome (Figure 5).

Soils are habitats for diverse microorganisms that are either harmful (pathogenic) or beneficial (symbiotic) to plants, comprising organic substances, water, and nutrients. Food production requires essential nutrients, metabolites, and water that the soil provides to plants [5,278,279]. Besides the nutrient resources in the soil, plants attract microbes via root exudates, root border cells, and mucilage formation that serve as food for the rhizosphere microbiome and their assembly [280]. Interestingly, soil contains an extensive reservoir of microorganisms (1 × 10^9^ microbial cells g^−1^ dry soil) and microbial diversity (1 × 10^5^ microbial species g^−1^ dry soil) [281,282]. Rhizospheric microbes are crucial in diverse biological processes, including soil structure, climate regulation, disease control, and organic matter decomposition. They also induce plant growth via nutrient uptake, cycling water availability, and the formation of stable aggregates to reduce the risks of soil erosion [283,284,285]. In addition, these microbes can improve plant health and serve as biocontrol agents against plant pathogens as well as biofertilizers (i.e., *Rhizobium*, *Azotobacter*, and *Bacillus subtilis*) [273,286]. Accordingly, elucidating the functional role of these microbial communities has garnered enormous efforts from botanical scientists over two decades to eliminate hunger and improve soil fertility by reducing synthetic fertilizers [287,288,289,290]. Fortunately, legume plants represent a solution, including fava beans, soybeans, lentils, and cowpea. They recruit symbiotic *Rhizobia* from the soil into nodules BNF. BNF is an eco-friendly and manageable strategy by harnessing nitrogen-fixing endophytic and free-living rhizobacteria to increase N levels in agricultural land by converting N_2_ to the fixed form, NH_3_ [291]. The biologically fixed amounts of atmospheric N into NH_3_ inside root nodules by legumes account for about 65% of N utilization in global agriculture via N-fixing *Rhizobia* [292]. Accordingly, this process allows legume plants to grow well in N-deficient soils, eliminating N fertilizer application [293]. The inoculation of *Rhizobia* strains (native or commercial) was studied on the growth and nodulation of three cowpea (*Vigna unguiculata*) genotypes in semiarid regions of Kenya. Field trials were performed in a randomized complete block design with three replications, and the symbiotic efficiency (SE) of *Rhizobium* isolates was evaluated. In the field, *Rhizobia* inoculation significantly increased nodulation and shoot DW compared to the uninoculated controls. Interestingly, *Rhizobia* inoculation significantly increased yields, whereas inoculation with native isolates recorded a 22.7% increase compared to uninoculated control in the first season and a 28.6% increase in yield in the second season. This study concluded that the efficient native *Rhizobia* in smallholder farms has promising potential to improve cowpea yield under a changing climate. In particular, *R. tropici* clone H53, *Mesorhizobium* sp. WSM3874, and *R. pusense* strain Nak353 showed more superiority in all the tested parameters [294]. A recent study revealed that plant exudates play a role in the development of bacterial biofilms (multicellular communities of microorganisms) in the soil, and creating microaerophilic conditions improved the nitrogen-fixing efficacy of diazotrophic bacteria in the soil [295] and exhibited enhanced flavonoid compounds. This study also hypothesized that manipulating flavonoid synthetic pathways leads to the induction of BNF in cereals through biofilm formation in soil diazotrophs [296,297]. N-limitation induces the production of exopolysaccharides (EPS), which may function as a barrier to block excess oxygen in the air and provide a suitable microaerobic condition for bacteria inside the EPS to fix N [298,299]. A large amount of EPS was reported in response to N limitation by cellulolytic bacteria (e.g., *Bacillus*, *Pedobacter*, *Chryseobacterium*, and *Flavobacterium* [300].

Applying the symbiotic association between *Rhizobia* and leguminous plants is a significant research area to reduce the dependency on chemical fertilizers. *Bradyrhizobium*, *Azospirillum*, and *Rhizobium* had high plant-growth-promoting capacity and N-fixing efficiency, indicating abundant functional microbial resources in extreme soil environments [301,302]. On the other hand, inoculation of *Azospirillum*, *Burkholderia*, and other N-fixing bacteria had little success in the field [303]. Thus, developing a compatible host for the indigenous N-fixing bacteria is critical for colonizing the host and improving nitrogenase activity.

## 9. Genomic Approaches for Improving N-Fixation

The identification of new genes involved in N fixation, investigation of the genome organization of N-fixing species, and the discovery of novel diazotrophic species have all been made possible by plant and microbial genomics. The primary genes responsible for N fixation encode nitrogenase enzymes, which play a crucial role in converting N_2_ into NH_4_^+^ [304,305]. These enzymes are classified into several types based on their structural, evolutionary, and functional characteristics. The first three forms of nitrogenase share similarities in their architecture, evolutionary origins, and mechanisms. However, they differ in the type of metal they use as a cofactor, either vanadium or molybdenum, or they may rely solely on iron as their cofactor. In contrast, the fourth type of nitrogenase operates differently; it utilizes a molybdopterin-containing system and depends on the presence of superoxide. This distinct form of nitrogenase was initially identified in a specific species called *Streptomyces thermoautotrophicus*. The complete genomic sequence of *S. thermoautotrophicus* and the precise amino acid sequence of this fourth type of nitrogenase have not been determined [306,307]. The fact that the inventory and distribution of species encoding for the fourth kind of nitrogenase are unknown highlights how few diazotrophs’ genomes have been sequenced. *A. vinelandii* is one of the extensively researched diazotroph with sequenced genomes that can fix N in an aerobic culture. It has served as a model for N-fixation for decades due to several factors, including (i) its genetic adaptability; (ii) its capacity to fix N in aerobic growth conditions; (iii) its nutritional flexibility, which is demonstrated by its capacity to fix N via three different pathways; and (iv) its capacity to adapt its metabolism to a variety of nutrients and media additives; and (v) the genome’s comprehensive sequencing and manual editing [308,309]. *A. vinelandii* has a single circular chromosome with 5,365,318 base pairs in its genome, which is expected to code for 5051 proteins [309]. The diazotroph *Pseudomonas stutzeri* is the closest companion of *A. vinelandii* with a sequenced genome [310]. Moreover, *P. stutzeri* grows aerobically and can only fix N in microaerobic conditions [311,312]. Alternatively, in ambient oxygen concentrations (20% O_2_), *A. vinelandii* can catalyze N fixation in an oxygen-sensitive process [313,314]. Its respiratory protection system makes it feasible to undertake two incompatible biological processes, oxidative phosphorylation and N fixation, concurrently. *A. vinelandii* may modify its respiration rate to maintain a low amount of cytoplasmic oxygen while N-fixing is taking place [315]. Five terminal oxidases were found in the genome, along with additional NADH oxidoreductases and other respiratory complexes that provide electrons to terminal oxidases, raising oxygen consumption [309]. Some of these genes, such as cydAB I, have been linked to respiratory protection and are crucial for aerobic N-fixation [306,307]. The discovery of novel N-fixing organisms and the quantity and kind of nitrogenases coding in these species may also be targeted by genome sequencing utilizing NifD sequences as queries [316,317,318]. The genomes of diazotrophs also encode for other proteins comparable to NifD in addition to the Nif/Vnf/AnfD sequences [318].

## 10. Conclusions and Future Perspectives

N is an essential macronutrient required for the growth and development of all living organisms. N uptake, transport, assimilation, and metabolism are finely tuned in plants. Additionally, N serves as a key mediator in plant–microbe interactions, fostering beneficial relationships through symbiotic partnerships that enhance nutrient availability. Moreover, N plays a pivotal role in a plant’s response to various environmental stresses, such as drought, salinity, and disease. Understanding N dynamics is critical for optimizing crop yields and sustainability, as it impacts the health and productivity of individual plants and the entire ecosystem and food security.

Consequently, harnessing N in agriculture addresses the challenges of feeding a growing global population while minimizing the environmental footprint of farming practices. In this sense, improving NUE by plants is a promising strategy to reduce the negative impact of using synthetic fertilizers. BNF is an eco-friendly and manageable strategy to increase N levels in agricultural land. Thus, improving BNF by genome editing could replace synthetic fertilizers to reduce dependency on chemical fertilizers. In addition, discovering new plant microbiomes and understanding their structure, abundance, and diversity will open a new avenue to better understanding the agricultural scenario to sustain global food security under limited N conditions. Despite understanding the role of N in plant biology, the precise molecular mechanisms governing N sensing in plants are still unclear, which could lead to the development of more efficient N-use strategies.

Furthermore, investigating the effects of climate change on N availability and its environmental impact becomes increasingly crucial. For example, arctic land areas are expected to warm, resulting in significant permafrost thawing [319,320] and significant alterations in ecological functioning [321]. With thawing, a massive pool of immobile C trapped in permafrost [322] becomes available for breakdown and remobilization, resulting in CO_2_ and methane (CH_4_) emissions. The gaseous C leak from thawing permafrost is being extensively researched in order to determine the extent of the permafrost–carbon feedback to the climate [323,324]. Permafrost soils, on the other hand, are huge N reservoirs, with an approximate amount of 67 billion tons of total N in the top 3 m [325]. Organically bound N is mineralized upon thawing, resulting in the release of NH_4_^+^ and NO_3_^−^ and driving nitrification and denitrification, the two primary processes producing greenhouse gas in soils [326], and these microorganisms can convert to harmful NO or N_2_ gas [327]. In addition to methane and carbon dioxide fluxes, warming arctic soils release N, which can increase N_2_O emissions, another potent greenhouse gas. The significance and amount of N_2_O released from arctic soils are unknown. There is a crucial need for a combination of in situ field investigations, laboratory experiments, and cutting-edge metagenomic techniques to better understand the role of microorganisms in climate change and seasonal changes in bacteria.

## Figures and Tables

**Figure 1 biomolecules-13-01443-f001:**
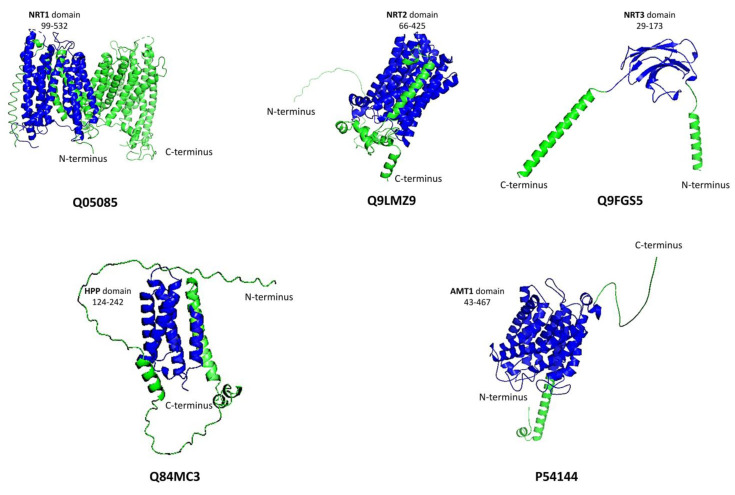
Secondary structure of N transporter families in higher plants. Except for NRT1, which is represented by an experimentally solved 3D structure (PDB: 4OH3), the other four transporters are represented by their alpha fold models. Blue regions refer to the nitrogen transporter domains, and green regions refer to the rest of the protein regions.

**Figure 2 biomolecules-13-01443-f002:**
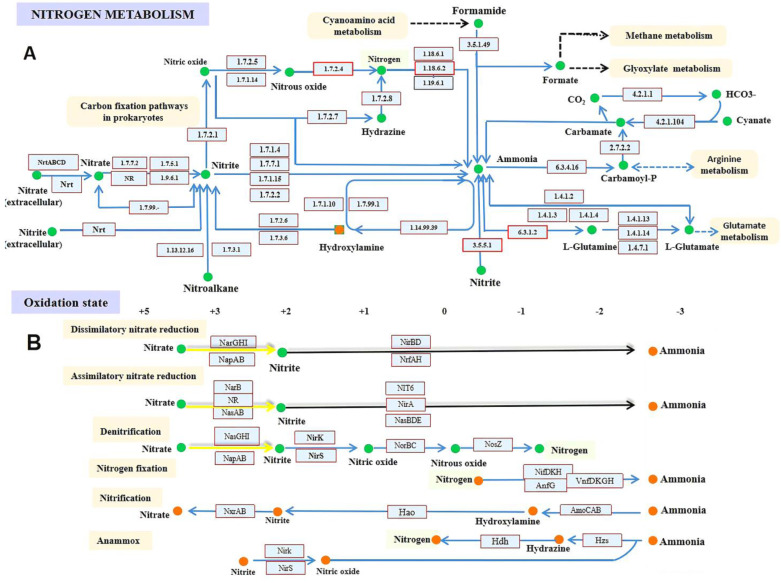
Overview of N metabolism pathways. (**A**). An illustration shows all the genes, enzymes, and substrates defined in the KEGG Pathway database regarding N metabolism. (**B**). The oxidation state comprises six sub-pathways: N fixation, assimilatory NO_3_^−^ reduction, dissimilatory NO_3_^−^ reduction, denitrification, nitrification, complete nitrification, comammox, and anammox.

**Figure 3 biomolecules-13-01443-f003:**
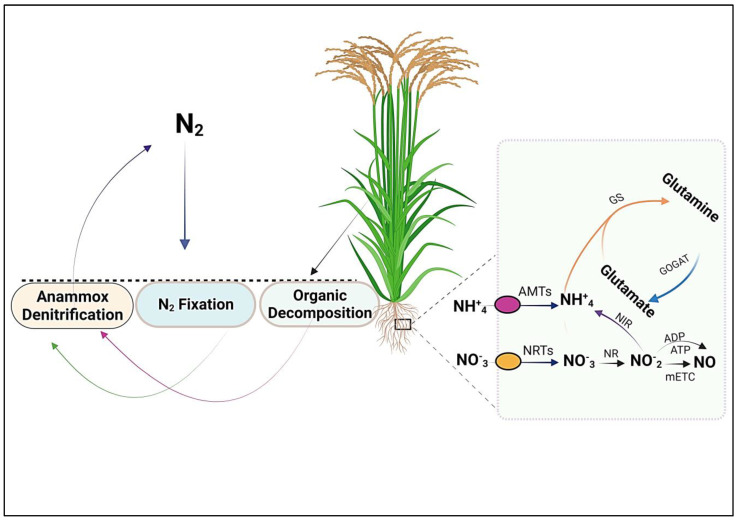
The N cycle, uptake, and assimilation. The main sources of soil N are N fixation, synthetic fertilizers, and organic decomposition. N uptake in the form of NO_3_^−^ or NH_4_^+^ through NRTs and AMTs, respectively. GS catalyzes glutamine synthesis from glutamate and NH_4_^+^ ions. At the same time, GOGAT catalyzes the synthesis of glutamate from glutamine and α-ketoglutarate, and thus, GS and GOGAT play central roles in the N metabolism and assimilation in plants.

**Figure 4 biomolecules-13-01443-f004:**
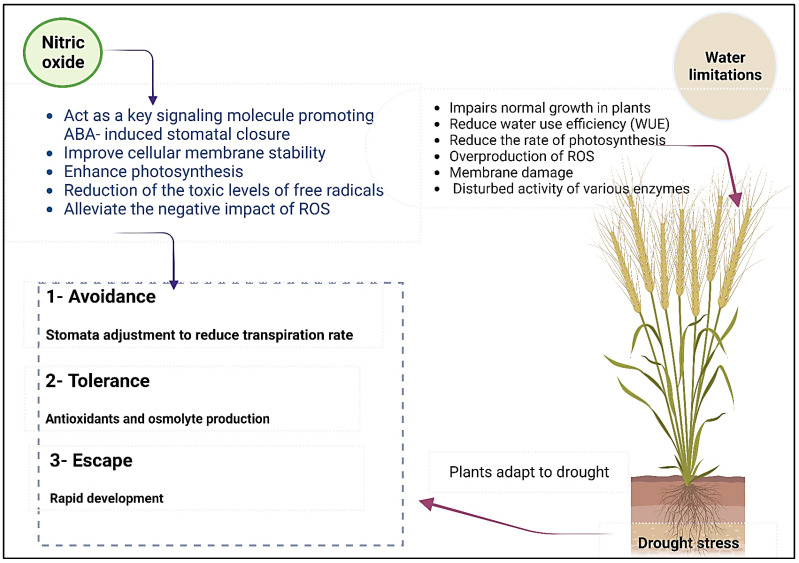
Adverse effects of drought stress on plant growth led to membrane damage, overproduction of ROS, and the deleterious effect on enzyme activity and other physiological processes such as photosynthesis rate. The induction of defense strategies in plants by NO through activation of antioxidant enzymes reduces the toxicity of free radicals and ROS negative impact and improves photosynthesis and membrane stability, which finally helps plants to avoid, tolerate, or escape drought conditions.

**Figure 5 biomolecules-13-01443-f005:**
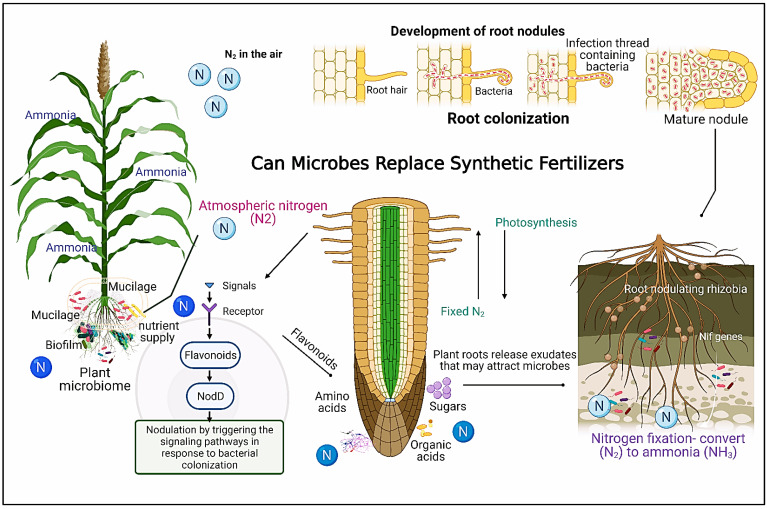
A simplified model represents the microbial beneficial effects and the signal pathways involved in modulating the effects on N availability. This overview of plant–microbe interactions in the rhizosphere represents the bacterial colonization of the roots to enhance nutrient uptake, resulting in plant growth promotion. N source limitation also increases NH_4_^+^ and NO_3_^−^ uptake activity as a promising strategy to achieve the sustainability of agriculture. In response to the nutrient deficiency, plant roots release some exudates that may recruit beneficial microbes to improve nutrient uptake. In return, they make nutrients available to plants that influence plant development. The availability of N is one of the critical elements in the process of BNF that converts atmospheric N_2_ to NH_3_. Accordingly, this model confirms that microbes might replace synthetic fertilizers in the future.

**Table 1 biomolecules-13-01443-t001:** List of genes and enzymes involved in the nitrogen cycle.

No.	KEGG ID	Gene Symbol	Gene Name	EC Number	Reference
1	K04561	NORB	Nitric oxide reductase subunit B	1.7.2.5	[125,126]
2	K15877	CYP55	Fungal nitric oxide reductase	1.7.1.14	[131,132]
3	K00376	NOSZ	Nitrous-oxide reductase	1.7.2.4	[133,134,135]
4	K02586	NIFD	Nitrogenase molybdenum-iron protein alpha chain	1.18.6.1	[136]
5	K22896	VNFD	Vanadium-dependent nitrogenase alpha chain	1.18.6.2	[137]
6	R05186	NIFF	Nitrogenase	1.19.6.1	[138]
7	K01455	FORMAMIDASE	Formamidase	3.5.1.49	[139]
8	K20935	HDH	Hydrazine dehydrogenase	1.7.2.8	[140]
9	K20932	K20932	Hydrazine synthase subunit	1.7.2.7	[141]
10	K01672	CA	Carbonic anhydrase	4.2.1.1	[142,143]
11	K01725	CYNS	Cyanate lyase	4.2.1.104	[144]
12	K00368	NIRK	Nitrite reductase (NO-forming)	1.7.2.1	[145]
13	K02575	NRT2, NARK, NRTP, NASA	MFS transporter, NNP family, nitrate/nitrite transporter	------	[146]
14	K15576	NRTA, NRTB, NRTC, NASD	Nitrate/nitrite transport system substrate-binding protein	7.3.2.4	[147]
15	K00370	NARG, NARZ, NXRA	Nitrate reductase/nitrite oxidoreductase, alpha subunit	1.7.5.1 1.7.99.-	[148]
16	K10534	NR	Nitrate reductase (NAD(P)H)	1.7.1.1 1.7.1.2 1.7.1.3	[149]
17	K00367	NARB	Ferredoxin-nitrate reductase	1.7.7.2	[150]
18	K02567	NAPA	Nitrate reductase (cytochrome)	1.9.6.1	[130]
19	K17877	NIT-6	Nitrite reductase (NAD(P)H)	1.7.1.4	[128]
20	K00366	NIRA	Ferredoxin-nitrite reductase	1.7.7.1	[151]
21	K00362	NIRB	Nitrite reductase (NADH) large subunit	1.7.1.15	[152]
22	K03385	NRFA	Nitrite reductase (cytochrome c-552)	1.7.2.2	[153]
23	K00459	NCD2, NPD	Nitronate monooxygenase	1.13.12.16	[154]
24	K19823	NAO	Nitroalkane oxidase	1.7.3.1	[155]
25	R00143		Hydroxylamine reductase (NADH)	1.7.1.10	[156]
26	K05601	HCP	Hydroxylamine reductase	1.7.99.1	[157]
27	K10535	HAO	Hydroxylamine dehydrogenase	1.7.2.6	[157]
28	R10230		Hydroxylamine oxidase (cytochrome)	1.7.3.6	[158]
29	K10944	PMOA-AMOA	Methane/ammonia monooxygenase subunit A	1.14.18.3 1.14.99.39	[159]
30	K00926	ARCC	Carbamate kinase	2.7.2.2	[160]
31	K01948	CPS1	Carbamoyl-phosphate synthase (ammonia)	6.3.4.16	[161]
32	K00260	GUDB, ROCG	Glutamate dehydrogenase	1.4.1.2	[162]
33	K00261	GLUD1_2, GDHA	Glutamate dehydrogenase (NAD(P)+)	1.4.1.3	[163]
34	K00262	E1.4.1.4, GDHA	Glutamate dehydrogenase (NADP+)	1.4.1.4	[164]
35	K01915	GLNA, GLUL	Glutamine synthetase	6.3.1.2	[165]
36	K01501	E3.5.5.1	Nitrilase	3.5.5.1	[166]
37	K00265	GLTB	Glutamate synthase (NADPH) large chain	1.4.1.13	[167]
38	K00264	GLT1	Glutamate synthase (NADH)	1.4.1.14	[168]
39	K00284	GLU, GLTS	Glutamate synthase (ferredoxin)	1.4.7.1	[169]
40	K00372	NASC, NASA	Assimilatory nitrate reductase catalytic subunit	1.7.99.-	[127]
41	K26139	NASD, NASB	Nitrite reductase [NAD(P)H] large subunit	1.7.1.4	[127]
42	K15864	NIRS	Nitrite reductase (NO-forming)/hydroxylamine reductase	1.7.2.1 1.7.99.1	[170,171]
43	K00531	ANFG	Nitrogenase delta subunit	1.18.6.1	[120]
44	K04561	NORB	Nitric oxide reductase subunit B	1.7.2.5	[125,126]

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
