# Peer review of "Nitrogen Journey in Plants: From Uptake to Metabolism, Stress Response, and Microbe Interaction"

_biomolecules, 2023, doi:10.3390/biom13101443_

Round 1

Reviewer 1 Report

This paper displayed a long story of nitrogen Journey in Plants. However, the content of the article seems a little redundant, and I suggest the structures of this paper can be improved as follows. 1.Introduction part: More words should be putted in the N functions in plants. 2.Nitrogen uptake: N bioavailability in rhizosphere (including microbe’s function)-uptake by transporters in roots (including NRT, AMTs and others?)-N metabolism in plants (especially to present how the N is translocated from roots to shoots)- functions of N signals (NO) in mitigating salt stress- the effect of N input on environment and how to management.

Author Response

Response to Reviewer 1 Comments

Comments and Suggestions for Authors

This paper displayed a long story of nitrogen Journey in Plants. However, the content of the article seems a little redundant, and I suggest the structures of this paper can be improved as follows. 1.Introduction part: More words should be putted in the N functions in plants. 2.Nitrogen uptake: N bioavailability in rhizosphere (including microbe’s function)-uptake by transporters in roots (including NRT, AMTs and others?)-N metabolism in plants (especially to present how the N is translocated from roots to shoots)- functions of N signals (NO) in mitigating salt stress- the effect of N input on environment and how to management.

Response: Two paragraphs have been inserted into the introduction to clarify the importance of N as an essential macronutrient for plant growth and development, as well as its effects on plant adaptation to abiotic stress factors.

Reviewer 2 Report

Dear Authors,

This review aims to cover too many things and, in my opinion, it doesn't do so correctly. It seems as if it's two different papers that have been attempted to be merged to create something new. But, in my understanding, it's not done with the necessary attention, depth, and care, and furthermore, significant crucial points are not addressed at all throughout the paper. It aims to cover three major areas: the entire nitrogen metabolism, the role of nitrogen in plant stress responses, and the interactions of microbiota in terms of nitrogen fixation with plants. Too many different things to make them fit well, something that in my opinion the authors haven't achieved with this version.

In my opinion, the article should be completely reworked, adding critical information and removing sections that do not address the interrelation of topics with sufficient depth. Additionally, it seems to me that they are not indicated or justified in terms of why they are included.

Majors:

In my understanding, the paper exhibits a significant bias towards the GS/GOGAT cycle, neglecting and not citing the equally or perhaps more important role of nitrate reductase in nitrogen assimilation. To my understanding, this should be addressed by the authors.

Another key point is that microalgae are not mentioned at all, despite their crucial role that has been studied in nitrogen metabolism in microalgae to understand the nitrogen metabolism in higher plants. This is especially true for the transport systems of ammonium and nitrate, as well as the pivotal role of nitrate reductase.

In my opinion, the paragraph dedicated to the introduction is too short and doesn't fulfill its purpose adequately. It falls short in introducing the reader to most of the topics discussed in the review. It only mentions a couple of them. I recommend that the authors expand it or consider removing it and incorporating its information into the following paragraphs.

Each of the paragraphs and sub-paragraphs should have a numbering.

L66: Plant nitrate transporters (NRTs) section: 1-Comment on what the literature says about the transcriptional regulation of these transporters. 2-It is not mentioned that nitrite can be transported by any of the nitrate transporters. Please comment on this fact. What role do the nitrite transporters play? Are they only specific ones?

NRT Structural analysis and AMTs Structural analysis sections, I don't see what Figure 1 contributes, and I also don't believe that these sections have enough substance to stand on their own. Each of them can be combined with the previous section.

In all the figures, the captions do not provide enough information to fully understand what they indicate or contribute. An example of this is Figure 2; there are two clearly differentiated parts, but they are not labeled in the caption. Please, number each one as A and B.

Nitrogen metabolism pathway section.  This section is very confusing. The paper is about nitrogen metabolism in plants; however, in this section, there is no distinction made between genes and pathways of plants versus those that only exist in microorganisms. It describes all of them together. It needs to be rewritten, clearly distinguishing between pathways in plants and microorganisms.

L167: from 3??? Do you mean -3?

Note that Table 1 would go in the supplementary material.

L190-200: In my opinion, a lot of information that I don't see contributing to the paper. What does it add?

L175: “Overview of nitrogen metabolism pathways” It would be more appropriate to use "inorganic nitrogen," as organic nitrogen is not discussed.

L215-L218: Key point, before these enzymes act, the nitrate reductase has been demonstrated to be a critical player in the regulation of nitrogen assimilation and is not discussed in the paper.

L219: “The process of nitrogen uptake by plants from the soil for growth has been well studied. Plants acquire nitrogen  in either ammonium or nitrate” Repeated, it has been mentioned before.

L220; “Nitrate is converted into ammonium,” A sentence too short to express the importance of nitrate reductase in plants, as I've mentioned before, it is crucial and the role of nitrate reductase should be further emphasized in the paper.

L273: “The pivotal function of GS within the nitrogen metabolism pathway of plants has been extensively investigated” I agree with that, but the Nitrate Reductase also has it, if its role in this paper is not discussed, crucial information is omitted.

Fig. 3: This figure is too large for the information it actually contributes to the paper, which has already been written in the paper.

L302: “adding N ” What sources? Please indicated

The section, Nitric oxide enhances plant survival under drought stress: Significant point, there is no mention of the way in which nitric oxide is synthesized in plants, in which nitrate reductase plays a fundamental role alongside the molybdoenzyme mARC.

L419: “Nitrous oxide and methane lead to greenhouse gas (GHG) emissions and reduce food quality” I can't find the sense of why this sentence is here. Please, provide reasoning and cite why nitrous oxide and methane are being discussed here.

L423: “methemoglobinemia” Explain in more detail what it consists of and what it's about.

L425: “In Argentina, the national food law set the limit for human consumption at 11.3 mg L−1 NO3 − -N” a reference is needed

L443:” In addition, the expensive cost, environmental hazards, soil fertility reduction, and other negative consequences” This sentence or reasoning is unfinished. As I see it, each part has been written by a different author, evident from the distinct way of expression in each section. This section needs a grammar and style revision in English.

L463: “In addition, N supply is necessary to provide N required for nucleotide and protein biosynthesis with other functions, i.e., carbon fixation capacity. Redundant, this idea has been expressed before.

L510: “Considering the human–microbe interaction to find non-pathogenic microbes (especially in the rhizobia genomes) that can colonize their host plant in field conditions” I don't understand what the human microbiota has to do with this. I don't understand this idea. Please rephrase the sentence to help  better comprehend what you are trying to express.

L538: “In addition, metabolites secreted by plant roots, such as flavonoids, benzaldehyde, biochanin A, salicylic acid, terpenoids, and tryptophan.” Another example of an unfinished or poorly expressed idea or sentence, it's not clear what the authors are trying to convey. Please, rephrase it or remove it. In my understanding, another example that the authors have not taken due care when submitting the final manuscript version.

L543: “For example, N-fixer microbes convert nitrogen  from the atmosphere into reactive nitrogen that plants can use in the natural rhizosphere as an alternative to chemical fertilization practices” Another idea that has already been expressed before and, at this point in the manuscript, doesn't make sense to cite. The same thing happens again in this sentence L545: “ Nitrogen is required to synthesize chlorophyll and other amino acids …” It gives the impression that none of the authors has read the entirety of all sections thoroughly and thoughtfully. This leaves a very negative impression. A critical point is for the authors to review the entire paper and avoid these types of redundancies and inconsistencies.

L555: “Biological nitrogen fixation converts dinitrogen (N2) to the fixed form of ammonia (NH3)” Another example of multiple redundancies, and furthermore, just after defining what BNF is: I won't point out any more. The authors must review everything and eliminate them.

L563: “Nitrogen (N2)” Really, at this point!

L626: ” which can increase  nitrous oxide (N2O) emissions, another potent greenhouse gas” But they just said it a few sentences ago, please, don't repeat it again.

L614: “Arctic N cycling microbiota” But if it is not discussion about  the microbiota in this section, I don't understand the purpose of this section, I don't see what it contributes.

L639: “we also evaluate methods for genome modification”. Reading what they present next, I don't understand why they claim to have evaluated something. In reality, there is  not evaluation; it is just cite what other authors have done

Minors:

L70: Please defined NPF

L84: transceptor, Please define what this is.

L86: CIPK, Please define what this is.

L121: Please denined TMHs

L154: “comprises two structural copies” What are you trying to convey by saying they are dimeric?

L206: “genes” or enzymes?

L399:  nitrogen (N),” defined before

L444; “N use efficiency (NUE)” defined before

L459: ” g in vitro” Italic.

Several times, G”o”GAT, Typo

L238: “electron contributor” or electron donor?

Author Response

Response to Reviewer 2 Comments

This review aims to cover too many things and, in my opinion, it doesn't do so correctly. It seems as if it's two different papers that have been attempted to be merged to create something new. But, in my understanding, it's not done with the necessary attention, depth, and care, and furthermore, significant crucial points are not addressed at all throughout the paper. It aims to cover three major areas: the entire nitrogen metabolism, the role of nitrogen in plant stress responses, and the interactions of microbiota in terms of nitrogen fixation with plants. Too many different things to make them fit well, something that in my opinion the authors haven't achieved with this version.

In my opinion, the article should be completely reworked, adding critical information and removing sections that do not address the interrelation of topics with sufficient depth. Additionally, it seems to me that they are not indicated or justified in terms of why they are included.

Majors:

In my understanding, the paper exhibits a significant bias towards the GS/GOGAT cycle, neglecting and not citing the equally or perhaps more important role of nitrate reductase in nitrogen assimilation. To my understanding, this should be addressed by the authors.

Another key point is that microalgae are not mentioned at all, despite their crucial role that has been studied in nitrogen metabolism in microalgae to understand the nitrogen metabolism in higher plants. This is especially true for the transport systems of ammonium and nitrate, as well as the pivotal role of nitrate reductase.

In my opinion, the paragraph dedicated to the introduction is too short and doesn't fulfill its purpose adequately. It falls short in introducing the reader to most of the topics discussed in the review. It only mentions a couple of them. I recommend that the authors expand it or consider removing it and incorporating its information into the following paragraphs.

Each of the paragraphs and sub-paragraphs should have a numbering.

Point 1: L66: Plant nitrate transporters (NRTs) section: 1-Comment on what the literature says about the transcriptional regulation of these transporters. 2-It is not mentioned that nitrite can be transported by any of the nitrate transporters. Please comment on this fact. What role do the nitrite transporters play? Are they only specific ones?

Response:

1- A paragraph has been added about the regulation of nitrogen transporters (knockout or overexpression) by transcription factors (TFs) and their families

2- A paragraph has been inserted depicting the nitrite uptake and the genes involved in this process.

NRT Structural analysis and AMTs Structural analysis sections, I don't see what Figure 1 contributes, and I also don't believe that these sections have enough substance to stand on their own. Each of them can be combined with the previous section.

Response: In Figure 1, we try to visualize the 3D structure of nitrogen transporters, either nitrate or ammonium representatives. We modified it by adding the domain location at every structure in addition to coloring it with a distinguished color. We agree with the reviewer in merging the structural analysis of nitrogen transporters with their respective former sections.

Figure 1. Secondary structure of nitrogen plant transporter families. Except for NRT1, which is represented by an experimentally solved 3D structure (PDB: 4OH3), the other 3 transporters are represented by their alphafold models. Blue regions refer to the nitrogen transporter domains, and green regions refer to the rest of the protein regions.

In all the figures, the captions do not provide enough information to fully understand what they indicate or contribute. An example of this is Figure 2; there are two clearly differentiated parts, but they are not labeled in the caption. Please, number each one as A and B.

Response: We fixed the legend as requested, Figure 2: Overview of N metabolism pathways. A. An illustration shows all the genes, enzymes, and substrates defined in the KEGG Pathway database regarding N metabolism. B. Oxidation state comprises six sub-pathways: N fixation, Assimilatory NO3− reduction, Dissimilatory NO3− reduction, Denitrification, Nitrification, Complete nitrification, comammox, and Anammox.

Nitrogen metabolism pathway section.  This section is very confusing. The paper is about nitrogen metabolism in plants; however, in this section, there is no distinction made between genes and pathways of plants versus those that only exist in microorganisms. It describes all of them together. It needs to be rewritten, clearly distinguishing between pathways in plants and microorganisms.

Response 1: The goal is to fully explain the metabolic pathway for nitrogen synthesis in all living organisms in general and regardless of a specific organism because initially, the genes associated with nitrogen synthesis were identified and found in a few bacteria, and then the researchers discovered and identified many of these genes in many plants such as; Arabidopsis thaliana, Saccharum officinarum, and Medicago truncatula. Therefore, we have explained the path in general to provide information that supports this review article and to benefit future researchers interested in the field of nitrogen fixation.

Reference’s:

-D.R. Woods, S.J. Reid. Recent developments on the regulation and structure of glutamine synthetase enzymes from selected bacterial groups. Fems Microbiol. Rev., 11 (1993), pp. 273-283

-A. Theron, R.L. Roth, H. Hoppe, C. Parkinson, C.W.V.D. Westhuyzen, S. Stoychev, I. Wiid, R.D. Pietersen, B. Baker, C.P. Kenyon. Differential inhibition of adenylylated and deadenylylated forms of M. tuberculosis glutamine synthetase as a drug discovery platform. PLoS ONE, 12 (2017), Article e0185068

-R. Mathis, P. Gamas, Y. Meyer, J.V. Cullimore. The presence of GSI-like genes in higher plants: support for the paralogous evolution of GSI and GSII genes. J. Mol. Evol., 50 (2000), pp. 116-122

-J. Biesiadka, A.B. Legocki. Evolution of the glutamine synthetase gene in plants. Plant Sci., 128 (1997), pp. 51-58.

-S. Ghoshroy, M. Binder, A. Tartar, D.L. Robertson Molecular evolution of glutamine synthetase II: phylogenetic evidence of a non-endosymbiotic gene transfer event early in plant evolution. BMC Evol. Biol., 10 (2010), p. 198

Point 2:  L167: from 3??? Do you mean -3?

Note that Table 1 would go in the supplementary material.

Response 2: Yes, -3

Point 3: L190-200: In my opinion, a lot of information that I don't see contributing to the paper. What does it add?

Response 3: This part aims to clarify the role of the ammonia (NH3) compound that plays various roles in different metabolic pathways, and this highlights the importance of ammonia produced through biosynthesis through the genes mentioned.

Point 4: L175: “Overview of nitrogen metabolism pathways” It would be more appropriate to use "inorganic nitrogen," as organic nitrogen is not discussed.

Response 4: We are discussing inorganic nitrogen mainly in the nitrogen cycle. For example, ammonium (NH4+) and nitrates (NO3–) predominate in the soil's inorganic fraction. Because these are the primary form that plants can uptake for their needs. In addition, ammonium occurs in both exchangeable and non-exchangeable forms. The other forms of inorganic nitrogen in soil are nitrogen gas (N2) and nitrites (NO2–). On the other hand, organic nitrogen refers to any organic compound containing nitrogen that includes amino acids, proteins, nucleotides, etc., along with nitrogen bound to residues of decomposing plant and animal matter and humus.

Point 5: L215-L218: Key point, before these enzymes act, the nitrate reductase has been demonstrated to be a critical player in the regulation of nitrogen assimilation and is not discussed in the paper.

Response 5: We already discussed this part from line no. 210 to 216. Also, we already added a paragraph with an example to explain the role of this gene in various plants, please see line no. 216 to 220

“And the biological function and role of these genes have been illustrated in many plants, such as; during the Medicago truncatulaSinorhizobium meliloti Symbiosis (Medicago truncatula) Berger et al., 2020, induced lateral root development (Arabidopsis thaliana) Kolbert et al., 2008 and Managing Cotton Nitrogen Supply ( Gossypium species) Thomas et al., 1998.”

Reference’s:

-Berger A, Boscari A, Horta Araújo N, Maucourt M, Hanchi M, Bernillon S, Rolin D, Puppo A and Brouquisse R (2020) Plant Nitrate Reductases Regulate Nitric Oxide Production and Nitrogen-Fixing Metabolism During the Medicago truncatula–Sinorhizobium meliloti Symbiosis. Front. Plant Sci. 11:1313. doi: 10.3389/fpls.2020.01313

-Kolbert, Z., Bartha, B., and Erdei, L. (2008). Exogenous auxin-induced NO synthesis is nitrate reductase-associated in Arabidopsis thaliana root primordia. J. Plant Physiol. 165, 967–975. doi: 10.1016/j.jplph.2007.07.019

-Thomas J. Gerik, Derrick M. Oosterhuis, H. Allen Torbert, Managing Cotton Nitrogen Supply,Advances in Agronomy,Academic Press,64, 1998:115-147, https://doi.org/10.1016/S0065-2113(08)60503-9.

Point 6: L219: “The process of nitrogen uptake by plants from the soil for growth has been well studied. Plants acquire nitrogen  in either ammonium or nitrate” Repeated, it has been mentioned before.

Response 6: Deleted

Point 7: L220; “Nitrate is converted into ammonium,” A sentence too short to express the importance of nitrate reductase in plants, as I've mentioned before, it is crucial and the role of nitrate reductase should be further emphasized in the paper.

Response 7: We completely agree with your point and added NR's principal role in the text with tracked changes. We covered this part in the previous comment.

Point 8: L273: “The pivotal function of GS within the nitrogen metabolism pathway of plants has been extensively investigated” I agree with that, but the Nitrate Reductase also has it, if its role in this paper is not discussed, crucial information is omitted.

Response 8: Thanks for this valuable observation, and we totally agree with your suggestion of adding the functional role of the NO. It was added in the main text.

Point 9: Fig. 3: This figure is too large for the information it actually contributes to the paper, which has already been written in the paper.

Response 9:

Point 10: L302: “adding N ” What sources? Please indicated

Response 10:   (NH4NO3)

Point 11: The section, Nitric oxide enhances plant survival under drought stress: Significant point, there is no mention of the way in which nitric oxide is synthesized in plants, in which nitrate reductase plays a fundamental role alongside the molybdoenzyme mARC.

Response 11: We fixed this in the revised version by adding an additional paragraph.

Like all organisms, plants must respond to various extreme environmental cues by responding to various internal signals. Nitric oxide (NO) is a small lipophilic molecule that diffuses through plant cell membranes as active signals to thrive and survive under those conditions. In addition, NO is a relevant signal molecule in many plant processes. Several pathways for NO production, either oxidative in the presence of O2, or reduction from nitrite, have been described in plants.

  • KOLBERT, Zsuzsanna, et al. Gasotransmitters in action: nitric oxide-ethylene crosstalk during plant growth and abiotic stress responses. Antioxidants, 2019, 8.6: 167.‏

Also, NO regulates gene expression, modulates enzyme activities, and acts as a metabolic intermediate in energy regeneration. However, the mechanisms and proteins responsible for its synthesis remain complex, with many unresolved questions. Nitrate, its primary substrate, is required for signaling and is widely distributed in diverse tissues in plants. Nitrate reductase (NR) is a key enzyme for nitrogen acquisition by plants, algae, and fungi and has been proposed as a crucial enzymatic source of nitric oxide (NO). Moreover, several studies showed the involvement of NO in reproductive processes, control of development, and the regulation of physiological responses such as stomatal closure. NO also regulates the expression of several genes involved in the synthesis of and response to pathogen attacks as well as reproductive mechanisms that operate during pollen recognition by the stigma.

  • Wilson, Ian D., Steven J. Neill, and John T. Hancock. "Nitric oxide synthesis and signalling in plants." Plant, cell & environment5 (2008): 622-631.‏-
  • Chamizo-Ampudia, Alejandro, et al. "Nitrate reductase regulates plant nitric oxide homeostasis." Trends in Plant Science2 (2017): 163-174.‏
  • NEILL, Steven J.; DESIKAN, Radhika; HANCOCK, John T. Nitric oxide signalling in plants. New Phytologist, 2003, 159.1: 11-35.‏
  • Cao, Yangrong, et al. "Extracellular ATP is a central signaling molecule in plant stress responses." Current opinion in plant biology20 (2014): 82-87.‏

In addition, NR is about 200 KDa and contains two subunits, each bearing three prosthetic groups: FAD, heme b557, and molybdenum. In an NR subunit, molybdenum is bound to a tricyclic pyranopterin and chelated by a dithiolene named the molybdenum cofactor (Moco). Amongst the reducing sources of NO in plants, nitrite can be reduced to NO by a family of five molybdenum (Mo) containing enzymes, which include nitrate reductase, xanthine oxidase reductase (XOR), aldehyde oxidase (AO), sulfite oxidase (SO), and mARC.

Recent findings assumed that most plants' main enzyme responsible for NO production is the molybdoenzyme NR.

  • Berger, Antoine, et al. "Nitrate reductases and hemoglobins control nitrogen-fixing symbiosis by regulating nitric oxide accumulation." Journal of Experimental Botany 72.3 (2021): 873-884.

In a study by Berger et al., 2020, three NR genes (MtNR1, MtNR2, and MtNR3) were discovered in the genome of Medicago truncatula and addressed their expression, activity, and potential involvement in NO production during the symbiosis between M. truncatula and Sinorhizobium meliloti. These results revealed that MtNR1 and MtNR2 gene expression and activity are correlated with NO production throughout the symbiotic process and that MtNR1 is mainly involved in NO production in mature nodules.

  • Berger, Antoine, et al. "Plant nitrate reductases regulate nitric oxide production and nitrogen-fixing metabolism during the Medicago truncatula–Sinorhizobium meliloti symbiosis." Frontiers in plant science11 (2020): 535004.‏

Interestingly, another study exhibited that several plant species released NO in the presence of nitrate in the soil. However, when plants were grown in soil containing ammonium, the functional role of NR was eliminated.

  • Wildt, J., Kley, D., Rockel, A., Rockel, P., & Segschneider, H. J. (1997). Emission of NO from several higher plant species. Journal of Geophysical Research: Atmospheres102(D5), 5919-5927.‏

Point 12: L419: “Nitrous oxide and methane lead to greenhouse gas (GHG) emissions and reduce food quality” I can't find the sense of why this sentence is here. Please, provide reasoning and cite why nitrous oxide and methane are being discussed here.

Response 12:  Removed

Point 13: L423: “methemoglobinemia” Explain in more detail what it consists of and what it's about.

Response 13: Fixed, Oxygen debt in the blood, a fatal illness known as methemoglobinemia or blue baby syndrome in infants under six months of age and some young animals

Point 14: L425: “In Argentina, the national food law set the limit for human consumption at 11.3 mg L−1 NO3 − -N” a reference is needed 

Response 14: https://doi.org/10.1016/j.agwat.2008.06.003

Point 15: L443:” In addition, the expensive cost, environmental hazards, soil fertility reduction, and other negative consequences” This sentence or reasoning is unfinished. As I see it, each part has been written by a different author, evident from the distinct way of expression in each section. This section needs a grammar and style revision in English.

Response 15: Removed

Point 16: L463: “In addition, N supply is necessary to provide N required for nucleotide and protein biosynthesis with other functions, i.e., carbon fixation capacity. Redundant, this idea has been expressed before.

Response 16: Removed

Point 17: L510: “Considering the human–microbe interaction to find non-pathogenic microbes (especially in the rhizobia genomes) that can colonize their host plant in field conditions” I don't understand what the human microbiota has to do with this. I don't understand this idea. Please rephrase the sentence to help  better comprehend what you are trying to express.

Response 17: Removed

Point 18: L538: “In addition, metabolites secreted by plant roots, such as flavonoids, benzaldehyde, biochanin A, salicylic acid, terpenoids, and tryptophan.” Another example of an unfinished or poorly expressed idea or sentence, it's not clear what the authors are trying to convey. Please, rephrase it or remove it. In my understanding, another example that the authors have not taken due care when submitting the final manuscript version.

Response 18: Fixed

Point 19: L543: “For example, N-fixer microbes convert nitrogen  from the atmosphere into reactive nitrogen that plants can use in the natural rhizosphere as an alternative to chemical fertilization practices” Another idea that has already been expressed before and, at this point in the manuscript, doesn't make sense to cite. The same thing happens again in this sentence L545: “ Nitrogen is required to synthesize chlorophyll and other amino acids …” It gives the impression that none of the authors has read the entirety of all sections thoroughly and thoughtfully. This leaves a very negative impression. A critical point is for the authors to review the entire paper and avoid these types of redundancies and inconsistencies.

Response 19: Removed

Point 20: L555: “Biological nitrogen fixation converts dinitrogen (N2) to the fixed form of ammonia (NH3)” Another example of multiple redundancies, and furthermore, just after defining what BNF is: I won't point out any more. The authors must review everything and eliminate them.

Response 20: Fixed

Point 21: L563: “Nitrogen (N2)” Really, at this point!

Response 21: Fixed

Point 22: L626: ” which can increase nitrous oxide (N2O) emissions, another potent greenhouse gas” But they just said it a few sentences ago, please, don't repeat it again.

Response 22: Fixed

Point 23: L614: “Arctic N cycling microbiota” But if it is not discussion about  the microbiota in this section, I don't understand the purpose of this section, I don't see what it contributes.

Response 23:  Fixed

Arctic permafrost is a fertile repository of Nitrogen. As a result, we tried to shed light on the cycling of the thawed permafrost and its N microbiota. Too much work regarding metagenomic analysis of Arctic N cycling microbiota is needed, so we merged this section with the conclusion part as a future perspective.

Point 24: L639: “we also evaluate methods for genome modification”. Reading what they present next, I don't understand why they claim to have evaluated something. In reality, there is  not evaluation; it is just cite what other authors have done

Response 24: Thank you for this comment; we have deleted that sentence in the main text to avoid the readers’ confusion.

Point 25: Minors:

L70: Please defined NPF Done

L84: transceptor, Please define what this is. Done

L86: CIPK, Please define what this is. Done

L121: Please denined TMHs Done

L154: “comprises two structural copies” What are you trying to convey by saying they are dimeric? Could not find it in the text

L206: “genes” or enzymes? Done

L399:  nitrogen (N),” defined before Done (Deleted)

L444; “N use efficiency (NUE)” defined before Done (Deleted)

L459: ” g in vitro” Italic. Done

Several times, G”o”GAT, Typo Done

L238: “electron contributor” or electron donor? Electron donor

Response 25: Thank you for all these comments; our response was made in the main text with tracked changes.

Round 2

Reviewer 2 Report

Dear Authors,

I believe the authors have appropriately addressed all of my suggestions and comments, and I accept the paper in its current version

Author Response

We really appreciate your comments and time.